# The association between prolonged SARS-CoV-2 symptoms and work outcomes

Arjun K. Venkatesh[1,2]*, Huihui Yu[2], Caitlin Malicki[1], Michael Gottlieb[3], Joann G. Elmore[4], Mandy J. Hill[5], Ahamed H. Idris[6], Juan Carlos C. Montoy[7], Kelli N. O'Laughlin[8], Kristin L. Rising[9,10], Kari A. Stephens[11,12], Erica S. Spatz[13,14‡], Robert A. Weinstein[15,16‡], for the INSPIRE Group[¶]

1 Department of Emergency Medicine, Yale School of Medicine, New Haven, Connecticut, United States of America, 2 Center for Outcomes Research and Evaluation (CORE), Section of Cardiovascular Medicine, Yale School of Medicine, Connecticut, United State of America, 3 Department of Emergency Medicine, Rush University Medical Center, Chicago, Illinois, United States of America, 4 Division of General Internal Medicine and Health Services Research, David Geffen School of Medicine at the University of California, Los Angeles (UCLA), Los Angeles, California, United States of America, 5 Department of Emergency Medicine, UTHealth Houston, Houston, Texas, United States of America, 6 Department of Emergency Medicine, University of Texas Southwestern Medical Center, Dallas, Texas, United States of America, 7 Department of Emergency Medicine, University of California, San Francisco, San Francisco, California, United States of America, 8 Departments of Emergency Medicine and Global Health, University of Washington, Seattle, Washington, United States of America, 9 Department of Emergency Medicine, Sidney Kimmel Medical College, Thomas Jefferson University, Philadelphia, Pennsylvania, United States of America, 10 Center for Connected Care, Thomas Jefferson University, Philadelphia, Pennsylvania, United States of America, 11 Department of Family Medicine, University of Washington, Seattle, Washington, United States of America, 12 Department of Biomedical Informatics and Medical Education, University of Washington, Seattle, Washington, United States of America, 13 Department of Epidemiology, Yale School of Public Health, New Haven, Connecticut, United States of America, 14 Section of Cardiovascular Medicine, Yale School of Medicine, New Haven, Connecticut, United States of America, 15 Division of Infectious Diseases, Department of Internal Medicine, Rush University Medical Center, Chicago, Illinois, United States of America, 16 Department of Medicine, Cook County Hospital, Chicago, Illinois, United States of America

‡ ESS and RAW are co-senior authors on this work.
¶ Membership of the INSPIRE Group is listed in the S1 Appendix.
* arjun.venkatesh@yale.edu

**Data Availability Statement:** Study data is owned and managed directly by the grant recipient (Rush University) and the funder (Centers for Disease Control and Prevention). We ask that you update

## Abstract

While the early effects of the COVID-19 pandemic on the United States labor market are well-established, less is known about the long-term impact of SARS-CoV-2 infection and Long COVID on employment. To address this gap, we analyzed self-reported data from a prospective, national cohort study to estimate the effects of SARS-CoV-2 symptoms at three months post-infection on missed workdays and return to work. The analysis included 2,939 adults in the Innovative Support for Patients with SARS-CoV-2 Infections Registry (INSPIRE) study who tested positive for their initial SARS-CoV-2 infection at the time of enrollment, were employed before the pandemic, and completed a baseline and three-month electronic survey. At three months post-infection, 40.8% of participants reported at least one SARS-CoV-2 symptom and 9.6% of participants reported five or more SARS-CoV-2 symptoms. When asked about missed work due to their SARS-CoV-2 infection at three months, 7.2% of participants reported missing ≥10 workdays and 13.9% of participants reported not returning to work since their infection. At three months, participants with ≥5 symptoms had a higher adjusted odds ratio of missing ≥10 workdays (2.96, 95% CI 1.81–4.83) and not returning to work (2.44, 95% CI 1.58–3.76) compared to those with no

the data availability statement as follows: The data underlying the results presented in the study are from the INSPIRE Registry. The coordinating center, Rush University, can be contacted via email at inspirepub@rush.edu to request information related to confidential data.

**Funding:** RAW. 75D30120C08008. Centers for Disease Control and Prevention, National Center of Immunization and Respiratory Diseases. https://www.cdc.gov/funding/index.html. The funder did not play any role in the study design, data collection and analysis, decision to publish or preparation of the manuscript.

**Competing interests:** I have read the journal's policy and the authors of this manuscript have the following competing interests: AHI receives research grant funding from the University of Texas Southwestern Medical Center outside of the submitted work. AKV receives grants from the Agency for Healthcare Research and Quality and the SAEM Foundation outside the submitted work. ESS receives grant funding from the National Heart, Lung, and Blood Institute (R01HL151240), and the Patient Centered Outcomes Research Institute (HM-2022C2-28354). JCCM receives research grant funding from SAMHSA (1H79TI084428-01 and 1H79TI085981-01, PI LeSaint), FDA (75F40122C00116, PI Anderson), NIH-NINDS (U24NS129501, PI Rodriguez) outside the submitted work. JE is Editor-in-chief of the Adult Primary Care topics at UpToDate. KLR receives research grant funding from Abbott Diagnostics, DermTech, MeMed, Prenosis, Siemens Healthcare Diagnostics, PROCOVAXED funded by NIAID 1R01AI166967, and PREVENT funded by CDC U01CK00048 outside the submitted work. KNO receives research grant funding for PROCOVAXED funded by NIAID R01 AI166967, PI: Rodriguez outside the submitted work. MG receives grant funding from the Biomedical Advanced Research and Development Authority Research Grant, the Bill and Melinda Gates Foundation, and the Society of Directors of Research in Medical Education Grant outside the submitted work. MJH receives research grant funding from an Investigator Award from Merck, MISP 100099, PI: Hill outside the submitted work. The following authors have declared that no competing interests exist: HY, KAS, RAW. This does not alter our adherence to PLOS ONE policies on sharing data and materials.

symptoms. Prolonged SARS-CoV-2 symptoms were common, affecting 4-in-10 participants at three-months post-infection, and were associated with increased odds of work loss, most pronounced among adults with ≥5 symptoms at three months. Despite the end of the federal Public Health Emergency for COVID-19 and efforts to "return to normal", policymakers must consider the clinical and economic implications of the COVID-19 pandemic on people's employment status and work absenteeism, particularly as data characterizing the numerous health and well-being impacts of Long COVID continue to emerge. Improved understanding of risk factors for lost work time may guide efforts to support people in returning to work.

## Introduction

The COVID-19 pandemic has resulted in tremendous economic dislocation in labor markets, creating historically volatile unemployment and reduced labor force participation rates due to unprecedented occupational health stresses and work loss [1]. Even as labor markets have stabilized in most countries including sustained periods of low unemployment [2], the economic impacts of the COVID-19 pandemic persist for many individuals as infections, hospitalizations and morbidity from severe acute respiratory syndrome coronavirus 2 (SARS-CoV-2) infection continue.

Most prior research has examined the macroeconomic employment effects of the pandemic and policy responses such as stay-at-home orders [3, 4], with less investigation of the direct relationship between SARS-CoV-2 illness and one's ability to work. One small retrospective study conducted early in the pandemic examining work outcomes among clinical cohorts reported that approximately half of individuals hospitalized with SARS-CoV-2 were unable to return to work six months after infection [5]. A more recent study using the United States (US) Current Population Survey found that work absences of up to one week due to acute SARS-CoV-2 illness were associated with less labor force participation and more work absences one year later, which is estimated to have reduced the US labor force by 500,000 people [6].

One possible mechanism for this long-term impact on employment is post-COVID conditions, which include a wide range of physical and mental health consequences that are present at least four weeks after SARS-CoV-2 infection [7, 8]. Post-COVID conditions, often referred to as Long COVID, affect nearly one-in-five adults with a history of SARS-CoV-2 infection [9] and may make returning to work or seeking employment more difficult [10]. An unadjusted analysis of an international convenience sample recruited via social media found that most individuals with SARS-CoV-2 symptoms beyond 28 days reported a reduced work schedule, suggesting a relationship between prolonged symptoms following an acute SARS-CoV-2 infection and short-term work loss [11]. However, the prevalence of Long COVID and its impact on work outcomes, such as return to work and missed workdays, is poorly understood [12].

To address this gap, we sought to utilize data from the Innovative Support for Patients with SARS-CoV-2 Infections Registry (INSPIRE) study to describe self-reported work outcomes related to acute SARS-CoV-2 infection and the presence of symptoms at three months post-infection.

## Materials and methods

### Study design

INSPIRE is a previously described prospective study designed to assess long-term symptoms and outcomes among persons with COVID-like illness who tested positive versus negative for SARS-CoV-2 at study enrollment [13]. Participants were enrolled virtually or in person

between December 7, 2020 and August 29, 2022 across eight study sites, including Rush University (Chicago, Illinois), Yale University (New Haven, Connecticut), the University of Washington (Seattle, Washington), Thomas Jefferson University (Philadelphia, Pennsylvania), the University of Texas Southwestern (Dallas, Texas), the University of Texas, Houston (Houston, Texas), the University of California, San Francisco (San Francisco, California) and the University of California, Los Angeles (Los Angeles, California). Inclusion criteria included age $\geq 18$ years, fluency in English or Spanish, self-reported symptoms suggestive of acute SARS-CoV-2 infection at time of testing (e.g., fever, cough), and testing for SARS-CoV-2 with an FDA-approved/authorized molecular or antigen-based assay within the preceding 42 days. Exclusion criteria included inability to provide consent, being lawfully imprisoned, inability of the study team to confirm the result of the index diagnostic test for SARS-CoV-2, having a previous SARS-CoV-2 infection >42 days before enrollment, and lacking access to an internet-connected device (e.g., smartphone, tablet, computer) for electronic survey completion. Participants with a positive SARS-CoV-2 test (COVID-positive) and a negative SARS-CoV-2 test (COVID-negative) were recruited in a 3:1 ratio. Participants completed a baseline survey and follow-up surveys every three months for up to 18 months post-enrollment, although only baseline and three-month follow-up surveys were included in this secondary analysis. The three-month survey was sent 76 days following enrollment, which was up to 118 days after the positive SARS-CoV-2 test, and participants had a 28-day window to complete the three-month survey. All study sites received institutional review board approval, including Rush University (IRB#20030902-IRB01), Yale University (IRB#2000027976), the University of Washington (IRB#STUDY00009920), Thomas Jefferson University (IRB##20P.1150), the University of Texas Southwestern (IRB#STU-202-1352), the University of Texas, Houston (IRB#HSC-MS-20-0981), the University of California, San Francisco (IRB#20–32222) and the University of California, Los Angeles (IRB#20–001683). Informed consent was obtained electronically for all study participants.

## Surveys

The baseline and three-month surveys included a variety of questions regarding sociodemographics, SARS-CoV-2 symptoms and overall health to assess long-term symptoms and outcomes related to COVID-like illness. To establish baseline employment status, the baseline survey asked, "Were you employed before the coronavirus outbreak?", with the following response options: Yes; No. To establish return to work status following a SARS-CoV-2 infection, the three-month survey asked, "Did you return to work after your COVID-19 like symptoms?", with the following response options: Yes, full-time; Yes, part-time or modified work; No; Not applicable. To establish workdays missed due to SARS-CoV-2 infection, the three-month survey asked, "Since before you had COVID-19 like symptoms, how many workdays or weeks did you miss because of health reasons?", with the following response options: I don't work; 0–5 workdays; 6–10 workdays; 10–20 workdays; up to 4 weeks.

Consistent with prior work [14–18], we administered the Centers for Disease Control and Prevention Persons Under Investigation symptom list to assess SARS-CoV-2 symptoms within both the baseline and three-month surveys, asking, "Do you currently have any of the following ongoing symptoms? (Select all that apply)", with the following response options: fever; feeling hot or feverish; chills; repeated shaking with chills; more tired than usual; muscle aches; joint pains; runny nose; sore throat; a new cough, or worsening of a chronic chough; shortness of breath; wheezing, pain or tightness in your chest; palpitations; nausea or vomiting; headache; hair loss; abdominal pain; diarrhea (>3 loose/looser than normal stools/24 hours); decreased smell or change in smell; decreased taste or change in taste; none of the above.

### Analysis

This analysis was restricted to COVID-positive participants who responded "yes" to being employed prior to the pandemic on the baseline survey and completed the three-month survey. To categorize missed workdays, we established a cutoff of greater than or equal to ten days based on the original design of survey response options.

For the primary analysis, we report unadjusted comparisons in baseline characteristics and outcomes using chi-square tests among groups with different numbers of symptoms (0, 1–2, 3–4, ≥5 symptoms). We examined the association of the number of symptoms at three-months with return to work and health-related work absenteeism (days missed ≥10) between enrollment and three-months using logistic regression models adjusting for age, gender, race, ethnicity, income, marital status, education, clinical comorbidity count, and SARS-CoV-2 variant time period [14]. Covariates were included in the model based on existing literature and unadjusted tests on the significance of association with outcomes.

For the secondary analysis, we presented the prevalence of individual symptoms and the distribution of symptom counts across work outcomes. As an exploratory analysis, we also conducted unadjusted comparisons to better understand whether work outcomes were associated with baseline annual income. All statistics and data analysis were performed in SAS 9.4.

## Results

### Survey completion

Among 8,950 individuals who completed informed consent, 6,075 were eligible for follow-up (Fig 1). A total of 4,588 participants completed the three-month survey, with survey completion rates varying slightly between the COVID-positive (77%) and COVID-negative (71%) groups. Among COVID-positive three-month survey respondents only (n = 3,533), we analyzed data from the 2,939 participants (83.2%) who responded "yes" to being employed prior to the pandemic on the baseline survey.

### Participant characteristics

The mean age was 40 years (SD 12.6), 64.1% were female, 69.5% were white, 61.2% were vaccinated for SARS-CoV-2 before index test, and 3.8% were hospitalized for SARS-CoV-2 infection. 1,732 (59.2%) and 282 (9.6%) reported 0 and ≥5 symptoms at three months, respectively (Table 1).

### Primary analysis

For missed workdays at three months post-infection, 197 participants (6.7%) missed ≥10 workdays, 2546 participants (87.0%) missed <10 workdays, 184 participants (6.3%) were not applicable (i.e. did not have work), and 1 participant (<1%) was missing a response. For return to work at three months post-infection, 386 participants (13.2%) did not return to work, 2388 participants (81.6%) returned to work, and 154 participants (5.3%) were missing responses.

The unadjusted bivariate analyses showed that participants with ≥5 symptoms had significantly higher odds of missing ≥10 workdays (18.8%) than participants with fewer (7.6–12.7%) or no symptoms (4.5%) at three months (p < .001). Similarly, the unadjusted bivariate analyses showed that participants with ≥5 symptoms were significantly more likely to not return to work (27.8%) compared to participants with fewer (15.1–16.3%) or no symptoms (10.5%) at three months (p < .001).

After adjusting for participants' sociodemographic and history of clinical conditions, participants with ≥5 symptoms had the greatest odds of missing ≥10 workdays and not returning

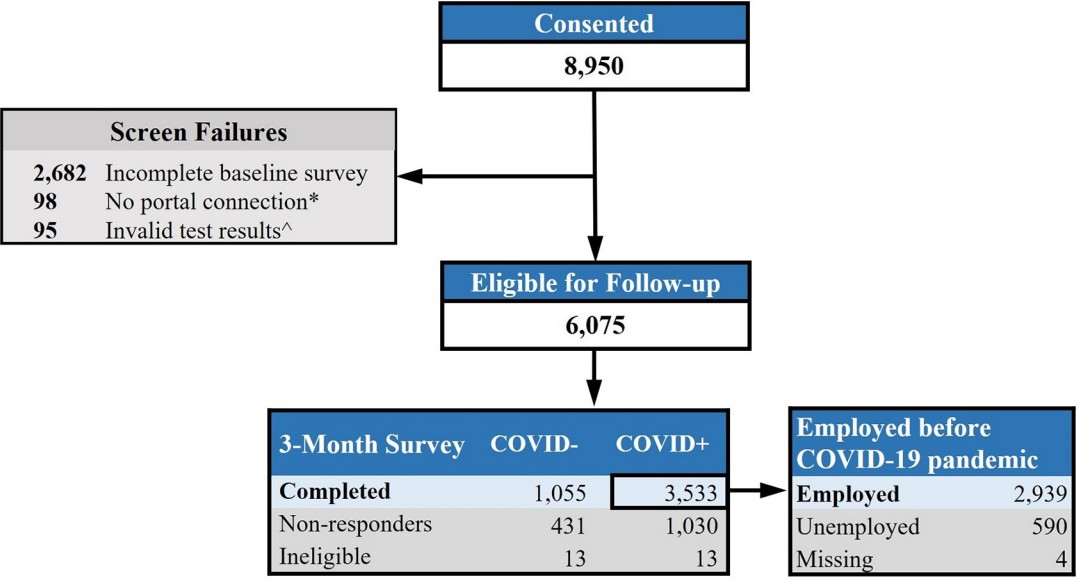

**Fig 1. INSPIRE participant flow diagram.** *No portal connection: Did not share medical records through electronic health portal, which was an eligibility requirement through 3/21/22. ^Invalid test results: Did not provide proof of SARS-CoV-2 test and/or had a positive SARS-CoV-2 test >42 days ago.

to work compared to those with fewer or no symptoms at three months (Fig 2). For participants with ≥10 missed workdays, there was a "dose-response" relation to number of symptoms: Compared to participants with no symptoms at three months, the odds of missing ≥10 workdays was two times higher in participants with 3–4 symptoms (adjusted odds ratio [aOR] = 1.94, 95% CI: 1.08–3.49) and nearly three times higher in participants with ≥5 symptoms (aOR = 2.96, 95% CI: 1.81–4.83). The difference in odds of missing ≥10 workdays was not significant among participants with 1–2 symptoms compared to those with no symptoms at three months (aOR = 1.58, 95% CI: 0.998–2.49). Compared to participants with no symptoms at three months, the odds of not returning to work were higher in participants with ≥5 symptoms (aOR = 2.44, 95% CI: 1.58–3.76) and in participants with 1–2 symptoms (aOR = 1.74, 95% CI: 1.23–2.47). The difference in odds of not returning to work was not significant among participants with 3–4 symptoms compared to those with no symptoms at three months (aOR = 1.25, 95% CI: 0.73–2.14).

## Secondary analysis

In secondary analyses of symptom burden, we found that participants who missed ≥10 workdays and participants who did not return to work reported a higher prevalence of each symptom at three months in comparison to those who missed <10 workdays and those who returned to work, respectively (Fig 3). We observed the five most prevalent symptoms among individuals missing ≥10 workdays and not returning to work were "more tired than usual", "headache", "muscle aches", "joint pains" and "shortness of breath", which also had the largest difference in symptom prevalence from those who experienced work loss and those who did not.

Additionally, when examining symptom count based on work outcomes, we found a statistically significant difference in the proportion of participants reporting higher numbers of symptoms for missing ≥10 workdays (p < .001) and not returning to work (p < .001) when compared to those missing <10 workdays and returning to work, respectively (Figs 2 and 4).

**Table 1. Baseline participant characteristics by number of symptoms at 3-months post-SARS-CoV-2 infection.**

| Characteristic[a]<br>n (%) | Total[b]<br>n = 2928 | Number of Symptoms at 3 months[b] (n, %) | | | | p-value |
|---|---|---|---|---|---|---|
| | | **0**<br>(1732, 59.2) | **1–2**<br>(677, 23.1) | **3–4**<br>(237, 8.1) | **≥5**<br>(282, 9.6) | |
| Age | | | | | | 0.306 |
| 18–34 | 1210 (41.3) | 750 (43.7) | 269 (40) | 87 (37) | 104 (37) | |
| 35–49 | 1019 (34.8) | 581 (33.8) | 249 (37.1) | 88 (37.4) | 101 (35.9) | |
| 50–64 | 548 (18.7) | 314 (18.3) | 121 (18) | 49 (20.9) | 64 (22.8) | |
| 65+ | 129 (4.4) | 73 (4.2) | 33 (4.9) | 11 (4.7) | 12 (4.3) | |
| Gender | | | | | | < .001 |
| Female | 1878 (64.1) | 1052 (62.5) | 458 (69.4) | 165 (72.7) | 203 (73.3) | |
| Male | 930 (31.8) | 612 (36.4) | 191 (28.9) | 58 (25.6) | 69 (24.9) | |
| Transgender/Non-Binary/Other | 39 (1.3) | 19 (1.1) | 11 (1.7) | 4 (1.8) | 5 (1.8) | |
| Ethnicity | | | | | | 0.024 |
| Non-Hispanic | 2485 (84.9) | 1494 (87.9) | 565 (85) | 190 (81.5) | 236 (85.5) | |
| Hispanic | 388 (13.3) | 205 (12.1) | 100 (15) | 43 (18.5) | 40 (14.5) | |
| Race | | | | | | < .001 |
| American Indian/Alaskan Native | 18 (0.6) | 11 (0.7) | 2 (0.3) | 1 (0.4) | 4 (1.4) | |
| Asian/Native Hawaiian/Pacific Islander | 389 (13.3) | 249 (14.7) | 92 (14) | 24 (10.5) | 24 (8.7) | |
| Black | 189 (6.5) | 103 (6.1) | 32 (4.9) | 26 (11.4) | 28 (10.1) | |
| White | 2036 (69.5) | 1217 (72) | 474 (72.3) | 148 (64.9) | 197 (71.1) | |
| Other | 220 (7.5) | 111 (6.6) | 56 (8.5) | 29 (12.7) | 24 (8.7) | |
| Education | | | | | | < .001 |
| Less than High school | 24 (0.8) | 11 (0.6) | 5 (0.8) | 4 (1.7) | 4 (1.4) | |
| High school graduate | 129 (4.4) | 64 (3.8) | 28 (4.2) | 20 (8.7) | 17 (6.1) | |
| Some College | 355 (12.1) | 185 (10.9) | 75 (11.3) | 31 (13.4) | 64 (22.9) | |
| 2-year degree | 189 (6.5) | 88 (5.2) | 47 (7.1) | 27 (11.7) | 27 (9.6) | |
| 4-year degree | 1018 (34.8) | 621 (36.6) | 228 (34.4) | 74 (32) | 95 (33.9) | |
| More than 4-year degree | 1156 (39.5) | 728 (42.9) | 280 (42.2) | 75 (32.5) | 73 (26.1) | |
| Marital Status | | | | | | < .001 |
| Married | 1665 (56.9) | 994 (57.4) | 397 (58.7) | 124 (52.8) | 150 (53.2) | |
| Divorced/Widowed/Separated | 253 (8.6) | 123 (7.1) | 58 (8.6) | 37 (15.7) | 35 (12.4) | |
| Never married | 1007 (34.4) | 615 (35.5) | 221 (32.7) | 74 (31.5) | 97 (34.4) | |
| Annual Family Income | | | | | | < .001 |
| <10,000 | 77 (2.6) | 45 (2.6) | 14 (2.1) | 5 (2.1) | 13 (4.6) | |
| 10,000–34,999 | 271 (9.3) | 137 (7.9) | 59 (8.7) | 36 (15.2) | 39 (13.8) | |
| 35,000–49,999 | 285 (9.7) | 132 (7.6) | 79 (11.7) | 31 (13.1) | 43 (15.2) | |
| 50,000–74,999 | 391 (13.4) | 217 (12.5) | 102 (15.1) | 32 (13.5) | 40 (14.2) | |
| 75,000+ | 1751 (59.8) | 1105 (63.8) | 393 (58.1) | 125 (52.7) | 128 (45.4) | |
| Prefer not to answer | 152 (5.2) | 96 (5.5) | 29 (4.3) | 8 (3.4) | 19 (6.7) | |
| Health Insurance | | | | | | < .001 |
| Private | 2378 (81.2) | 1460 (84.3) | 549 (81.1) | 177 (74.7) | 192 (68.1) | |
| Public | 383 (13.1) | 188 (10.9) | 89 (13.1) | 38 (16) | 68 (24.1) | |
| Private and public | 68 (2.3) | 40 (2.3) | 11 (1.6) | 11 (4.6) | 6 (2.1) | |
| None | 99 (3.4) | 44 (2.5) | 28 (4.1) | 11 (4.6) | 16 (5.7) | |
| Housing | | | | | | 0.450 |
| Unstable housing | 25 (0.9) | 12 (0.7) | 6 (0.9) | 4 (1.7) | 3 (1.1) | |
| Stable housing | 2898 (99) | 1719 (99.3) | 671 (99.1) | 233 (98.3) | 275 (98.9) | |
| Tobacco Use | | | | | | < .001 |

*(Continued)*

**Table 1.** (Continued)

| Characteristic[a]<br>n (%) | Total[b]<br>n = 2928 | Number of Symptoms at 3 months[b] (n, %) | | | | p-value |
|---|---|---|---|---|---|---|
| | | 0 | 1–2 | 3–4 | ≥5 | |
| | | (1732, 59.2) | (677, 23.1) | (237, 8.1) | (282, 9.6) | |
| Any tobacco use | 385 (13.1) | 221 (12.8) | 68 (10) | 40 (16.9) | 56 (19.8) | |
| No tobacco use | 2541 (86.8) | 1510 (87.2) | 609 (90) | 196 (83.1) | 226 (80.1) | |
| Comorbidities | | | | | | |
| Asthma | 331 (11.3) | 156 (9.1) | 78 (11.6) | 44 (19) | 53 (19.6) | < .001 |
| Hypertension | 330 (11.3) | 175 (10.2) | 72 (10.7) | 34 (14.7) | 49 (18.1) | < .001 |
| Diabetes | 118 (4) | 69 (4) | 27 (4) | 13 (5.6) | 9 (3.3) | 0.613 |
| Obesity | 763 (26.1) | 397 (23.2) | 179 (26.7) | 93 (40.3) | 94 (34.8) | < .001 |
| Emphysema/COPD | 47 (1.6) | 22 (1.3) | 16 (2.4) | 3 (1.3) | 6 (2.2) | 0.219 |
| Heart conditions | 28 (1) | 16 (0.9) | 6 (0.9) | 2 (0.9) | 4 (1.5) | 0.844 |
| Kidney disease | 6 (0.2) | 2 (0.1) | 1 (0.1) | 0 (0) | 3 (1.1) | 0.008 |
| Liver disease | 15 (0.5) | 4 (0.2) | 6 (0.9) | 2 (0.9) | 3 (1.1) | 0.075 |
| COVID-19 Testing Location | | | | | | < .001 |
| At home | 469 (16) | 312 (18.1) | 117 (17.3) | 19 (8) | 21 (7.5) | |
| Walk in clinic | 361 (12.3) | 198 (11.5) | 73 (10.8) | 34 (14.3) | 56 (20.1) | |
| Emergency department | 73 (2.5) | 29 (1.7) | 14 (2.1) | 11 (4.6) | 19 (6.8) | |
| Hospital | 218 (7.4) | 121 (7) | 51 (7.6) | 22 (9.3) | 24 (8.6) | |
| Drive up testing site | 1621 (55.4) | 974 (56.4) | 375 (55.6) | 131 (55.3) | 141 (50.5) | |
| Other | 177 (6) | 94 (5.4) | 45 (6.7) | 20 (8.4) | 18 (6.5) | |
| Hospitalization for SARS-CoV-2 | | | | | | < .001 |
| Not hospitalized | 2779 (94.9) | 1678 (97.7) | 641 (95.7) | 219 (94.8) | 241 (89.3) | |
| Hospitalized | 110 (3.8) | 40 (2.3) | 29 (4.3) | 12 (5.2) | 29 (10.7) | |
| Vaccination Status before SARS-CoV-2 Infection | | | | | | < .001 |
| Unvaccinated | 535 (18.3) | 288 (21.2) | 103 (19.7) | 46 (24) | 98 (38.4) | |
| Vaccinated | 1792 (61.2) | 1070 (78.8) | 419 (80.3) | 146 (76) | 157 (61.6) | |

COPD, chronic obstructive pulmonary disease.

[a]Participants missing characteristics were excluded from this table. Missing data was < 3% (n = 1–81) for all characteristics, except for Vaccination Status before SARS-CoV-2 Infection, where missing data was 20.5% (n = 601).

[b]Participants missing the number of symptoms at three months (n = 11) were excluded from this table.

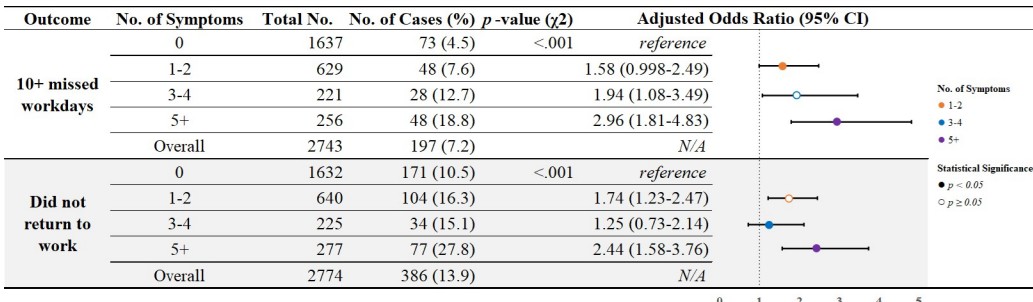

**Fig 2. Comparison of work outcomes stratified by the number of symptoms at 3-months post- SARS-CoV-2 infection.** CI, confidence interval. Figure excludes the following participants: Missing responses for symptoms at 3-month follow-up (n = 11); Missing or not applicable responses for "10+ workdays missed" (n = 185) and "Did not return to work" (n = 154). Logistic regression models adjusted for age, gender, race, ethnicity, income, marital status, education, clinical comorbidity count, and SARS-CoV-2 variant time period.

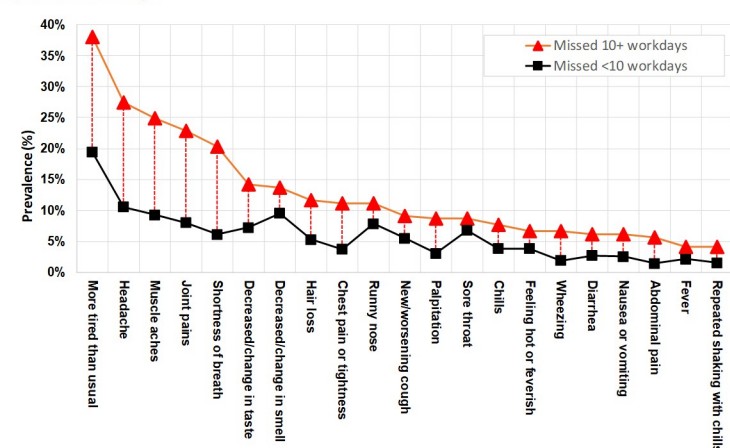

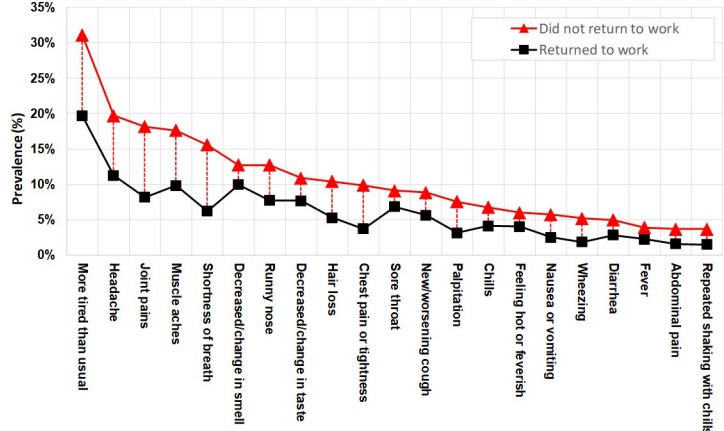

**Fig 3. Prevalence of individual symptoms at 3-months post-SARS-CoV-2 infection by missed workdays and return to work status.**

In an exploratory analysis of work outcomes based upon participant income strata, participants in the lowest income strata reported the highest proportions of work loss at three months both among participants not experiencing symptoms at three months as well as among participants with one or more symptoms at three months. Among participants who reported no symptoms at three months, there was a statistically significant difference in missed workdays ($p < .001$) and return to work status ($p = .002$) across income strata. Among those participants who reported one or more symptoms at three months, there was a statistically significant difference in missed workdays ($p < .001$) and return to work status ($p < .001$) across income strata. (Table 2).

## Discussion

The presence of SARS-CoV-2 symptoms three months after acute infection was pervasive and was associated with a greater likelihood of work loss in a prospective registry of adults with an initial SARS-CoV-2 infection. This association was especially pronounced among adults with a greater symptom burden ($\geq 5$ symptoms) at three months, who experienced two- to three-fold increased risk of substantial missed workdays and not returning to work compared to adults whose symptoms resolved by three months.

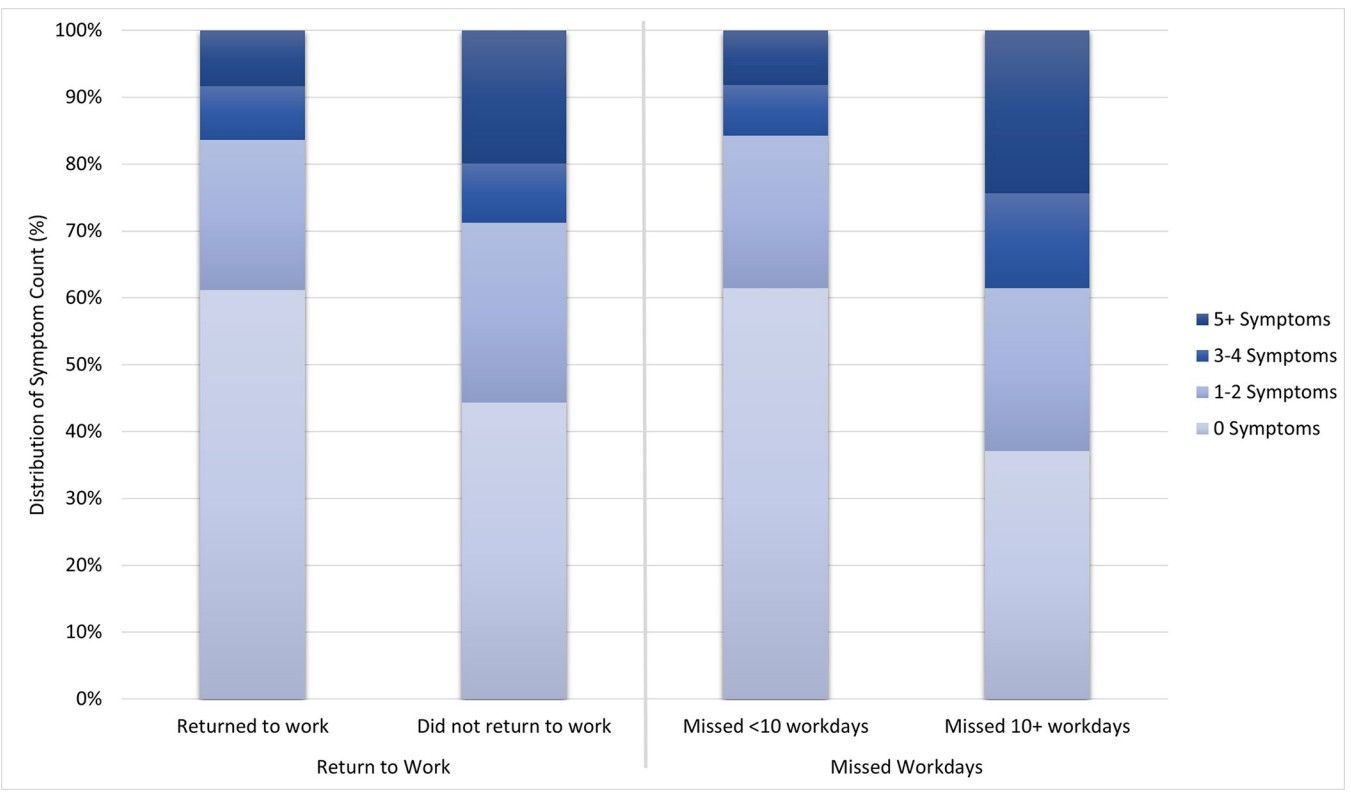

**Fig 4. Distribution of number of symptoms at 3-months post-SARS-CoV-2 infection by missed workdays and return to work status.**

Our work extends prior literature in a few notable ways. First, by capturing clinical information beyond the acute SARS-CoV-2 infection period and including primarily adults with milder disease, the relationships identified in the INSPIRE registry are likely more generalizable to working US adults than prior work in more limited populations. Second, because the INSPIRE registry includes baseline and longer-term follow up data, this analysis supports

**Table 2. Comparison of work loss by income strata among participants with no symptoms and one or more symptoms at 3-months post-SARS-CoV-2 infection.**

| Income Level | Event: Missed 10+ workdays[a] | | | | Event: Did not return to work[b] | | | |
|---|---|---|---|---|---|---|---|---|
| | 0 Symptoms* | | 1+ Symptoms* | | 0 Symptoms* | | 1+ Symptoms* | |
| | N = 1,637 | | N = 1,105 | | N = 1,632 | | N = 1,141 | |
| | N | Event N (%) | N | Event N (%) | N | Event N (%) | N | Event N (%) |
| < $10,000 | 34 | 8 (23.5%) | 24 | 6 (25.0%) | 39 | 14 (35.9%) | 29 | 12 (41.4%) |
| $10,000 - $34,999 | 123 | 14 (11.4%) | 119 | 20 (16.8%) | 131 | 32 (24.4%) | 128 | 49 (38.3%) |
| $35,000 - $49,999 | 124 | 11 (8.9%) | 141 | 18 (12.8%) | 126 | 12 (9.5%) | 146 | 30 (20.5%) |
| $50,000 - $74,999 | 211 | 8 (3.8%) | 157 | 11 (7.0%) | 211 | 15 (7.1%) | 163 | 24 (14.7%) |
| $75,000+ | 1,057 | 26 (2.5%) | 612 | 58 (9.5%) | 1,039 | 81 (7.8%) | 620 | 80 (12.9%) |
| Prefer not to answer | 88 | 6 (6.8%) | 52 | 11 (21.2%) | 86 | 17 (19.8%) | 55 | 20 (36.4%) |

[a]Participants missing a response for missed workdays at 3-month follow-up (n = 185) were excluded from this table.
[b]Participants missing a response for return to work at three months (n = 154) were excluded from this table.
* P < 0.005, indicating that statistically significant difference in each outcome by income category within a symptom strata.

temporally connecting persistent SARS-CoV-2 symptoms to long-term work outcomes. Third, the consistent relationship between higher three-month symptom burden and worse work outcomes further bolsters the likelihood that health effects of post-COVID conditions are sufficient to explain the well-documented relationship between health and work status. Lastly, the broad enrollment period of the INSPIRE registry from December 2020 through August 2022 demonstrates the durability of these findings despite the likely heterogeneous effects of SARS-CoV-2 infection upon work alongside evolving variants, severity of disease, vaccination and treatment.

Several mechanisms may support this identified relationship between Long COVID and work outcomes. First, evidence suggests SARS-CoV-2 can lead to a variety of negative health outcomes including the emergence of new chronic diseases, which can result in disabilities that may limit work ability [19, 20]. Second, prior work has shown a consistent relationship between Long COVID and depression and anxiety, both of which are closely linked to work participation and changes in work status [21]. Third, persistent fatigue and decreased exercise tolerance following SARS-CoV-2 infection may play a pivotal role in ability to return to work, particularly among those with physically demanding jobs and essential workers [22].

Given high SARS-CoV-2 vaccination rates (77% among participants with non-missing data) and the mild disease course observed in this cohort, the magnitude of work loss is striking. Extrapolating to 208 million adults working in the U.S., of whom at least 42% are estimated to have been infected with SARS-CoV-2 as of May 2022 [23], our data suggest that symptomatic SARS-CoV-2 infection may have contributed to over 12.9 million individuals not returning to work within three months of infection, of whom 2.4 million may have post-covid conditions. Given the disproportionate burden of SARS-CoV-2 infection observed among workers in public-facing industries, such as education and healthcare, the economic impacts of Long COVID may be more concentrated in select occupations and economic sectors [24].

Our finding of differential work outcomes based on income strata warrants further research. We found a stark but consistent relationship in which lower-income participants appear to have disproportionately experienced more work loss outcomes. Prior work has demonstrated that those of lower income are likely to be employed in occupations that are more sensitive to pandemic-related dislocations and often have less flexibility in work to accommodate conditions, such as Long COVID, with remote or hybrid work environment and adequate healthcare coverage, among other reasons [25, 26]. Given the complexity and intersectionality of income with race, ethnicity, and other social determinants of health, our work cannot imply causal relationships or explain these differences found between income groups but warrants future research that elucidates the potentially disproportionate impact of Long COVID on lower income workers.

As with all studies, this analysis has limitations. First, estimates may overattribute work loss due to SARS-CoV-2 as our analysis did not include COVID-negative participants and other pandemic-related causes. Second, our analyses considered each symptom equally, which may not capture nuanced relationships between symptom type, symptom severity and work outcomes. Third, survey questions and response options were developed iteratively in response to the changing landscape of the pandemic, and analysis was limited by survey design and available data. Fourth, there is some risk of recall bias in any survey-based study; however, several aspects of survey design should mitigate this risk including the use of questions stems to direct symptom questions to be COVID-related. Fifth, given limitations of sample size in our secondary analysis of work outcomes by income strata, findings were exploratory in nature as they are not based on a representative sample and do not include a sufficiently large sample in lower income groups to accommodate more granular analyses. Lastly, there is risk of residual

confounding as we were not able to adjust for all covariates, such as disability, which were found to be positively correlated with the covariates and outcomes.

## Conclusion

Findings suggest that prolonged SARS-CoV-2 symptoms are common, affecting 4 in 10 participants at three months post-infection and are associated with increased odds of work loss, with the most pronounced work loss association among adults with $\geq 5$ symptoms at three months. As data characterizing the numerous health and well-being impacts of Long COVID emerge, despite efforts to "return to normal," policymakers must consider the clinical and economic implications of the COVID-19 pandemic on people's employment status and work absenteeism and strategies for reducing absenteeism.

## Supporting information

**S1 Appendix. INSPIRE group.**
(DOCX)

## Acknowledgments

We would like to thank the California Department of Public Health, CTSI COVID Clinical Research Steering Committee, CTSI Office of Clinical Research Patient Navigation Team, and Bioinformatics Program Public Health Seattle King County for their assistance with participant recruitment. We would also like to thank the University of Washington Institute of Translational Health Sciences (ITHS) for support of the REDCap instance and for biomedical informatics resources used by the UW Clinical Core and Enrolling Site to enable study recruitment, which is funded by the National Center for Advancing Translational Sciences of the National Institutes of Health under award number UL1TR002319.

## Author Contributions

**Conceptualization:** Arjun K. Venkatesh.

**Data curation:** Arjun K. Venkatesh, Huihui Yu, Caitlin Malicki.

**Formal analysis:** Huihui Yu.

**Funding acquisition:** Michael Gottlieb, Robert A. Weinstein.

**Investigation:** Arjun K. Venkatesh, Michael Gottlieb, Joann G. Elmore, Mandy J. Hill, Ahamed H. Idris, Juan Carlos C. Montoy, Kelli N. O'Laughlin, Kristin L. Rising, Kari A. Stephens, Erica S. Spatz, Robert A. Weinstein.

**Methodology:** Arjun K. Venkatesh, Huihui Yu, Michael Gottlieb, Joann G. Elmore.

**Project administration:** Arjun K. Venkatesh, Caitlin Malicki, Michael Gottlieb, Joann G. Elmore, Mandy J. Hill, Ahamed H. Idris, Juan Carlos C. Montoy, Kelli N. O'Laughlin, Kristin L. Rising, Kari A. Stephens, Erica S. Spatz, Robert A. Weinstein.

**Supervision:** Michael Gottlieb, Joann G. Elmore, Kari A. Stephens, Robert A. Weinstein.

**Validation:** Huihui Yu.

**Visualization:** Huihui Yu, Caitlin Malicki.

**Writing – original draft:** Arjun K. Venkatesh, Huihui Yu, Caitlin Malicki.

**Writing – review & editing:** Arjun K. Venkatesh, Huihui Yu, Caitlin Malicki, Michael Gottlieb, Joann G. Elmore, Mandy J. Hill, Ahamed H. Idris, Juan Carlos C. Montoy, Kelli N. O'Laughlin, Kristin L. Rising, Kari A. Stephens, Erica S. Spatz, Robert A. Weinstein.

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
