## [Decision Letter · Decision Letter 0]

30 Apr 2024

PONE-D-24-09134The association between prolonged SARS-CoV-2 symptoms and work outcomesPLOS ONE

Dear Dr. Venkatesh,

Thank you for submitting your manuscript to PLOS ONE. After careful consideration, we feel that it has merit but does not fully meet PLOS ONE’s publication criteria as it currently stands. Therefore, we invite you to submit a revised version of the manuscript that addresses the points raised during the review process.

We look forward to receiving your revised manuscript.

Kind regards,

G. K. Balasubramani

Academic Editor

PLOS ONE

Journal Requirements:

2. Thank you for stating the following in the Competing Interests: 

   ""I have read the journal's policy and the authors of this manuscript have the following competing interests:

AHI receives research grant funding from the University of Texas Southwestern Medical Center during the conduct of the study and being a member of the Stryker Belfast Clinical Advisory Board outside the submitted work.

AV receives grants from the Agency for Healthcare Research and Quality and the SAEM Foundation outside the submitted work.

ESS receives grant funding from the National Heart, Lung, and Blood Institute (R01HL151240), and the Patient Centered Outcomes Research Institute (HM-2022C2-28354).

JCCM receives research grant funding from SAMHSA (1H79TI084428-01 and 1H79TI085981-01, PI LeSaint), FDA (75F40122C00116, PI Anderson), NIH-NINDS (U24NS129501, PI Rodriguez) outside the submitted work.

JE is Editor-in-chief of the Adult Primary Care topics at UpToDate. 

KLR receives research grant funding from Abbott Diagnostics, DermTech, MeMed, Prenosis, Siemens Healthcare Diagnostics, PROCOVAXED funded by NIAID 1R01AI166967, and PREVENT funded by CDC U01CK00048 outside the submitted work.

KNO receives research grant funding for PROCOVAXED funded by NIAID R01 AI166967, PI: Rodriguez outside the submitted work. 

MG receives grant funding from the Biomedical Advanced Research and Development Authority Research Grant, the Bill and Melinda Gates Foundation, and the Society of Directors of Research in Medical Education Grant outside the submitted work.

MJH receives research grant funding from an Investigator Award from Merck, MISP 100099, PI: Hill outside the submitted work. 

The following authors have declared that no competing interests exist: HY, KAS, RAW."

We note that one or more of the authors have an affiliation to the commercial funders of this research study : University of Texas Southwestern Medical Center

   "I have read the journal's policy and the authors of this manuscript have the following competing interests:

AHI receives research grant funding from the University of Texas Southwestern Medical Center during the conduct of the study and being a member of the Stryker Belfast Clinical Advisory Board outside the submitted work.

AV receives grants from the Agency for Healthcare Research and Quality and the SAEM Foundation outside the submitted work.

ESS receives grant funding from the National Heart, Lung, and Blood Institute (R01HL151240), and the Patient Centered Outcomes Research Institute (HM-2022C2-28354).

JCCM receives research grant funding from SAMHSA (1H79TI084428-01 and 1H79TI085981-01, PI LeSaint), FDA (75F40122C00116, PI Anderson), NIH-NINDS (U24NS129501, PI Rodriguez) outside the submitted work.

JE is Editor-in-chief of the Adult Primary Care topics at UpToDate. 

KLR receives research grant funding from Abbott Diagnostics, DermTech, MeMed, Prenosis, Siemens Healthcare Diagnostics, PROCOVAXED funded by NIAID 1R01AI166967, and PREVENT funded by CDC U01CK00048 outside the submitted work.

KNO receives research grant funding for PROCOVAXED funded by NIAID R01 AI166967, PI: Rodriguez outside the submitted work. 

MG receives grant funding from the Biomedical Advanced Research and Development Authority Research Grant, the Bill and Melinda Gates Foundation, and the Society of Directors of Research in Medical Education Grant outside the submitted work.

MJH receives research grant funding from an Investigator Award from Merck, MISP 100099, PI: Hill outside the submitted work. 

The following authors have declared that no competing interests exist: HY, KAS, RAW."

5.  One of the noted authors is a group or consortium "INSPIRE Group". In addition to naming the author group, please list the individual authors and affiliations within this group in the acknowledgments section of your manuscript. Please also indicate clearly a lead author for this group along with a contact email address.

Additional Editor Comments:

The manuscript received feedback from two reviewers, who have recommended major revisions to improve the paper's quality.

The study enrolled participants from eight different sites, but the data related to these sites has not been included in the table based on the number of symptoms. As such, it is unclear if the authors adjusted their model to account for any variation between these sites. It would be helpful to know the strategies the authors used to ensure that the results are not biased by differences between the sites.

One of the reviewers pointed out that using hair loss as an indicator of asymptomatic COVID-19 cases is not accurate. This, along with the classification of the number of symptoms category, raises questions about the validity of the analysis. The authors should consider alternative measures to accurately identify asymptomatic cases and reclassify the number of symptoms categories to ensure that the analysis is valid.

The study data showed that recall bias occurred, leading to biased estimates of associations between exposures and outcomes. It is important that the authors implement strategies to minimize the recall bias and enhance the validity and reliability of study findings. The authors should consider using objective measures, such as biomarkers or medical records, to validate self-reported data. Additionally, they should consider conducting a sensitivity analysis to determine the impact of recall bias on the study's findings.

Reviewers' comments:

Reviewer's Responses to Questions

**Comments to the Author**

1. Is the manuscript technically sound, and do the data support the conclusions?

Reviewer #1: Yes

Reviewer #2: Yes

2. Has the statistical analysis been performed appropriately and rigorously? 

Reviewer #1: Yes

Reviewer #2: Yes

3. Have the authors made all data underlying the findings in their manuscript fully available?

Reviewer #1: Yes

Reviewer #2: No

4. Is the manuscript presented in an intelligible fashion and written in standard English?

Reviewer #1: Yes

Reviewer #2: Yes

5. Review Comments to the Author

Reviewer #1: Venkatesh et al. investigated the link between prolonged SARS-CoV-2 symptoms and work outcomes in US. This analysis included 2939 participants and the results showed that 7.1% of participants reported missing ≥10 workdays and 13.9% of participants reported not back to work since their infection. Prolonged SARS-CoV-2 symptoms were linked with increased chances of work loss, most pronounced among adults with ≥5 symptoms at three months. The study highlights the effect of clinical implications of the COVID-19 pandemic on people’s employment status. The topic is interesting and important. Please see the comments below.

1. Vaccination status was included in the participant characteristics. Did the authors include the FDA-approved inhibitor, Paxlovid? That inhibitor was used as an antiviral.

2. The questions in the survey seem need to differentiate the mild symptoms and the severe symptoms as different symptoms have different effect. Simply count the numbers of symptoms may provide misleading results. What is the rational to include hair loss as one of the symptoms?

3. Participants enrolled into the survey are from eight universities in US. Participants from companies may also needed.

Reviewer #2: The authors conducted a prospective cohort study, the INSPIRE study, to assess the association between Long COVID symptoms and work outcomes. This study is informative and could meaningfully guide patients returning to work after COVID-19.

This manuscript is well-structured and comprehensive. Here are some comments to help improve the manuscript further:

(1) Line 126. It appears that the COVID-19 symptoms were self-reported by the participants. How did you mitigate recall bias, especially for COVID-19 symptoms that are easily confused with other conditions? If recall bias is acknowledged, it should be discussed in the limitations section of the manuscript.

(2) Line 146: Regarding the questions asked of participants, such as "COVID-19-like symptoms," how can participants without a medical background accurately identify COVID-19-like symptoms? Do you provide a list of symptoms, or is medical proof required from participants?

(3) Regarding the surveys, did you include questions about mental health disorders following COVID-19? Depression or anxiety could also prevent patients from returning to work, making it a relevant factor to consider in your study.

(4) Concerning the analysis results, did you investigate any potential interaction effects? Additionally, could you stratify the results based on high, middle, and low income levels? This would help determine whether the impact of COVID-19 symptoms to work varies across different income groups, which could provide more nuanced and informative insights.

(5) Which symptoms were most (or top 5?) significantly associated with affecting work outcomes? Additionally, did you find that vaccination had a protective effect that helped patients return to work earlier? These insights could be informative, and they would add considerable value to your study.

(6) In the discussion, it would be beneficial to explore the possible mechanisms or pathways through which Long COVID may affect work outcomes. One suggestion is to consider discussing the findings from recent retrospective cohort studies on Long COVID effects.

For example, COVID-19 may increase the risk of different kind of health outcomes (Bowe, B., Xie, Y. & Al-Aly, Z. Postacute sequelae of COVID-19 at 2 years. Nat Med 29, 2347–2357 (2023). https://doi.org/10.1038/s41591-023-02521-2).

Long COVID may increase the risk of depression and anxiety. (Zhang Y, Chinchilli VM, Ssentongo P, et al Association of Long COVID with mental health disorders: a retrospective cohort study using real-world data from the USABMJ Open 2024;14:e079267. doi: 10.1136/bmjopen-2023-079267)

Post-hospitalization COVID-19 may lead to disabilities and financial problems. (Admon AJ, Iwashyna TJ, Kamphuis LA, et al. Assessment of Symptom, Disability, and Financial Trajectories in Patients Hospitalized for COVID-19 at 6 Months. JAMA Netw Open. 2023;6(2):e2255795. doi:10.1001/jamanetworkopen.2022.55795)

Discussing these aspects and more references could make your discussion more comprehensive.

Overall, this is a great manuscript. Congratulations to the authors for their hard work.

6. PLOS authors have the option to publish the peer review history of their article (what does this mean?). If published, this will include your full peer review and any attached files.

Reviewer #1: No

Reviewer #2: No

---

## [Author Response · Author response to Decision Letter 0]

14 Jun 2024

Thank you for your comments and suggestions! As instructed, we have included a point-by-point response in the "Response to Reviewers" document uploaded with this submission.

---

## [Decision Letter · Decision Letter 1]

12 Jul 2024

The association between prolonged SARS-CoV-2 symptoms and work outcomes

PONE-D-24-09134R1

Dear Dr. Venkatesh,

We’re pleased to inform you that your manuscript has been judged scientifically suitable for publication and will be formally accepted for publication once it meets all outstanding technical requirements.

Kind regards,

G. K. Balasubramani

Academic Editor

PLOS ONE

Additional Editor Comments (optional):

Only a minor edit is needed for Table 1. The missing category in all the variables in Table 1 is unnecessary. The authors can add a footnote for Table 1 stating that the sum of certain variables may not equal the total number due to missing data. The significance reported in that table does not include the missing data, so there's no need for this to be included in the table.

Reviewers' comments:

Reviewer's Responses to Questions

**Comments to the Author**

1. If the authors have adequately addressed your comments raised in a previous round of review and you feel that this manuscript is now acceptable for publication, you may indicate that here to bypass the “Comments to the Author” section, enter your conflict of interest statement in the “Confidential to Editor” section, and submit your "Accept" recommendation.

Reviewer #2: All comments have been addressed

2. Is the manuscript technically sound, and do the data support the conclusions?

Reviewer #2: Yes

3. Has the statistical analysis been performed appropriately and rigorously? 

Reviewer #2: Yes

4. Have the authors made all data underlying the findings in their manuscript fully available?

Reviewer #2: Yes

5. Is the manuscript presented in an intelligible fashion and written in standard English?

Reviewer #2: Yes

6. Review Comments to the Author

Reviewer #2: Congratulations to the authors for this great manuscript. They solved my questions and concerns well.

7. PLOS authors have the option to publish the peer review history of their article (what does this mean?). If published, this will include your full peer review and any attached files.

Reviewer #2: No

---

## [Editor Report · Acceptance letter]

17 Jul 2024

PONE-D-24-09134R1 

PLOS ONE

Dear Dr. Venkatesh, 

I'm pleased to inform you that your manuscript has been deemed suitable for publication in PLOS ONE. Congratulations! Your manuscript is now being handed over to our production team.

Kind regards, 

on behalf of

Dr. G. K. Balasubramani 

Academic Editor

PLOS ONE